# Advances in HIV-1 Assembly

**DOI:** 10.3390/v14030478

**Published:** 2022-02-26

**Authors:** Grigoriy Lerner, Nicholas Weaver, Boris Anokhin, Paul Spearman

**Affiliations:** Division of Infectious Diseases, Cincinnati Children’s Hospital Medical Center and University of Cincinnati, 3333 Burnet Avenue, Cincinnati, OH 45229, USA; grigoriy.lerner@cchmc.org (G.L.); nicholas.weaver@cchmc.org (N.W.); boris.anokhin@cchmc.org (B.A.)

**Keywords:** RNA packaging, Gag protein, envelope glycoprotein, maturation, inositol hexakisphosphate, matrix protein, capsid protein, nucleocapsid protein, Env trafficking

## Abstract

The assembly of HIV-1 particles is a concerted and dynamic process that takes place on the plasma membrane of infected cells. An abundance of recent discoveries has advanced our understanding of the complex sequence of events leading to HIV-1 particle assembly, budding, and release. Structural studies have illuminated key features of assembly and maturation, including the dramatic structural transition that occurs between the immature Gag lattice and the formation of the mature viral capsid core. The critical role of inositol hexakisphosphate (IP6) in the assembly of both the immature and mature Gag lattice has been elucidated. The structural basis for selective packaging of genomic RNA into virions has been revealed. This review will provide an overview of the HIV-1 assembly process, with a focus on recent advances in the field, and will point out areas where questions remain that can benefit from future investigation.

## 1. Introduction

The assembly processes of enveloped viruses are inherently fascinating and complex. Discussions of viral lifecycles almost inevitably begin with the attachment and entry of a virus into a target cell. Perhaps these discussions should begin instead with a description of the creation of the particle (virus assembly). After all, what would attachment and entry mean in the absence of the interplay between virus and host that results in the formation of an infectious viral particle? From the standpoint of the virus, assembly is the central act of creation that determines all subsequent steps in viral replication. A defining characteristic of the assembly of all viruses, well-illustrated by HIV-1, is the intimate interaction between viral gene products and host cellular machinery. The assembly of HIV can be conceptualized as a well-choreographed series of virus–host cell interactions that have been driven largely by viral evolution and adaptation.

To introduce the major steps in HIV-1 assembly, a very brief overview will help to set the stage and is presented in Figure 1. Transcription of HIV-1 RNA is initiated from the integrated proviral DNA, producing a series of RNA products. The unspliced HIV-1 genomic RNA (gRNA) is exported from the nucleus and serves as the template for translation of Gag and Gag-Pol, which takes place on free cytosolic ribosomes. Gag is formed as a 55 kD precursor polyprotein, Pr55^Gag^, while the 160 kD Gag-Pol polyprotein is formed via a ribosomal frameshift occurring during translation at approximately 5% of the level of Pr55^Gag^. A subset of gRNA dimerizes and interacts with the nucleocapsid (NC) region of Gag, serving as the gRNA that is packaged into immature particles. Gag molecules interact with each other and with lipid microdomains on the plasma membrane (PM), generating the immature Gag lattice and inducing membrane curvature. Meanwhile, a singly spliced HIV-1 RNA is exported from the nucleus and becomes the mRNA template for Env (and Vpu) translation on ribosomes associated with the endoplasmic reticulum (ER). Env forms trimers in the ER, and progresses through the secretory pathway, where cleavage and glycosylation take place. Upon reaching the PM, Env is rapidly endocytosed through a clathrin/AP-2-dependent process. Intracellular Env is recycled to the PM, where it is specifically incorporated into developing particles in a manner regulated by its long cytoplasmic tail. Cleavage of Gag by the HIV-1 protease creates dramatic structural rearrangements and results in the formation of the mature, conical core of HIV, a process termed maturation that occurs simultaneously with budding. Mature infectious particles complete the budding process and are released from the cell, ready to infect a target CD4+ lymphocyte or macrophage. In sections below, we will expand on this overview, with an emphasis on recent developments along the pathways outlined in Figure 1.

This review will cover basic aspects of the assembly of HIV-1. Essential steps in particle assembly that have been known for decades will be summarized, so that more recent developments and areas requiring additional investigation can be presented clearly to the reader. We begin with the export of the viral genomic RNA and progress through discoveries relevant to the Gag protein and its cleavage products; the maturation process; Env synthesis, trafficking, and particle incorporation; and finally discuss host restriction factors that act along the assembly pathway.

## 2. HIV-1 Genomic RNA Nuclear Export and Trafficking

Transcription from the integrated proviral DNA genome produces a variety of RNA products, including the full-length 9 kb transcripts that perform a dual function: encoding the Gag and Gag-Pol proteins and serving as the gRNA for packaging and replication. Approximately one-half of the full-length HIV-1 RNA transcripts produced remain unspliced. Splicing of the full-length transcript results in more than 50 relevant transcripts that are either incompletely spliced (4 kb class) transcripts encoding Env, Vif, Vpr, and Vpu; or completely spliced transcripts (1.8 Kb class) encoding viral accessory proteins Tat, Rev, and Nef [1,2]. The completely spliced RNAs exit the nucleus via the canonical mRNA export pathway. The intron-containing unspliced and partially spliced RNAs require the HIV-1 Rev protein for their nuclear export. Rev binds to the Rev response element (RRE) of the RNA, located within the Env coding region, and directs nuclear export of the RNA through a CRM1/RanGTP-dependent export pathway [3]. The Mason-Pfizer monkey virus (MPMV) genome does not utilize a trans-acting adaptor such as Rev, but instead contains a structural element at its 3′ end termed the constitutive transport element (CTE) that mediates nuclear export [4]. The MPMV CTE interacts with host export proteins Nuclear RNA Export Factor 1 (NXF1/NXT1), which binds to TREX2 at the nuclear pore to promote nuclear export [5,6]. A logical question arising from use of these distinct nuclear export pathways is whether they influence subsequent events in the virus lifecycle, including subcellular localization of gRNA, translation of structural proteins, or RNA packaging. Such a difference in RNA localization could potentially have profound effects on assembly, and explain why HIV-1 may have evolved to utilize a different RNA export pathway to support assembly at the PM. In support of this idea, one report found that by replacing the RRE in HIV-1 gRNA with multiple copies of the CTE, the subcellular location of the RNA was altered from a diffuse cytoplasmic pattern to one that clustered near the centrosome, suggesting that the mode of nuclear exit indeed determines the subcellular fate of the RNA [7]. However, Chen and colleagues readdressed this question by comparing the distribution of HIV-1 RNAs exported through the CRM1 or NSF1 pathway using live cell imaging [8]. They found that CTE-containing HIV-1 RNA and RRE-containing HIV-1 RNA exhibited a similar cytoplasmic distribution and did not observe clustering near the centrosome. Single particle tracking revealed that HIV-1 RNAs expressed in the absence of Gag are transported within the cytoplasm predominantly through diffusion. This finding agrees with previous work showing that HIV-1 gRNA travels via diffusion throughout the cytoplasm, rather than by active transport [9,10]. Thus, it remains unclear what advantage the CRM1 export pathway offers as compared with NSF1-dependent nuclear export for HIV-1 genomic RNA.

## 3. RNA Dimerization and Packaging

The HIV-1 gRNA serves both as a template for translation of Gag and Gag-Pol proteins and as the genome for packaging into assembling virions. The RNA packaging or “psi” signal is a highly structured stretch of 150–250 bases near the 5′ end of the gRNA [11]. Stem-loop 3 (SL3) within psi binds to NC through a high-affinity interaction between the SL3 GGAG tetraloop and the NC zinc fingers, while additional interactions mediate NC binding to stem-loop 2 (SL2) [12,13]. In the context of full-length Gag, however, a purine-rich loop in SL1 was found to be the major binding site on gRNA [14,15]. Additional elements within the 5′ untranslated region (UTR) may also contribute to gRNA packaging, including the 5′ polyA hairpin [16,17]. Dimerization of gRNA is mediated by conserved elements within the 5′ UTR of gRNA [18,19,20], in particular the dimer initiation signal (DIS) within SL1 [21,22,23]. Encapsidation of two copies of RNA into developing particles is essential for viral recombination, so dimerization clearly is an advantage for the evolution of retroviruses. Recombination occurs frequently during the reverse transcription process as the reverse transcriptase (RT) enzyme switches between RNA templates, resulting in a hybrid proviral DNA copy [24,25,26]. Until recently, however, it was not known whether packaging of two RNAs was important for viral replication. Rawson and colleagues addressed this question by blocking homologous recombination in defined regions of the genome and examining progeny virions. Remarkably, blocking recombination led to a high rate of progeny with damaged or incomplete proviruses, indicating that recombination not only contributes to viral evolution, but is required for efficient HIV-1 replication and for maintaining the integrity of the genome [27]. The question arises, therefore, what is it that determines dimerization (and packaging) of the gRNA, versus use of gRNA as an mRNA template for translation? Recent studies indicate that the transcriptional start site for gRNA determines the fate of the RNA through its influence on the structure of the capped leader RNA [28,29,30]. The HIV-1 promoter includes three sequential guanosines at the U3-R junction that can serve as the site of initiation of transcription, producing gRNAs differing in one or two guanosines at the 5′ end. gRNAs with only a single guanosine (1G form) at the 5′ end are selected for packaging, while those with two or three guanosines (2G, 3G forms) serve as mRNA for translation. Furthermore, the Cap1G RNA forms dimers, while G2 and G3 forms remain monomeric. The structural basis for the fate of the gRNA was demonstrated by deuterium-edited NMR [28]. In the 1G form, the 5′,5′-triphosphate-linked 7-methylguanosine cap is hidden within a multi-hairpin structure that promotes dimerization, and in this form the cap is sequestered, inhibiting interactions with eukaryotic translation initiation factor 4E. The 1G structure also sequesters the major splice donor site and the translational start site, while exposing Gag binding sites. In contrast, the 2G and 3G transcripts maintain an accessible cap and exposed splice donor, allowing interaction with translation factors or splicing machinery, while inhibiting dimerization and sequestering the Gag binding sites. Thus, the 1G form creates dimeric gRNA that interacts with Gag for packaging, while the 2G and 3G gRNAs interact with cellular machinery directing splicing or translation. This remarkable discovery is illustrated in Figure 2. This work solves the previous mystery of how packaged HIV-1 gRNAs are selected from a pool of gRNA. Additional aspects of the psi structure and its contribution to HIV-1 assembly are discussed in the section on NC below.

## 4. Central Role of Gag in HIV-1 Assembly

The HIV-1 Gag protein is the master organizer of the virus assembly process. Gag is translated on cytosolic ribosomes as the precursor Pr55^Gag^, which forms the structural shell of the developing immature particle. Expression of Gag alone in a variety of cell types results in the formation of immature virus-like particles that bud from the PM [31,32,33]. An important concept in understanding the role of Gag versus its cleavage products is that immature (uncleaved) Gag performs essential functions relevant to the assembly process, including packaging of the gRNA, interactions with the PM, and multimerization leading to membrane curvature and budding; while rearrangements occurring in the mature virion are required in order to perform critical entry and post-entry events in the viral lifecycle. During or shortly following budding, the HIV protease (PR) cleaves Gag into its subunit components matrix (MA), capsid (CA), spacer peptide 1 (SP1), nucleocapsid (NC), spacer peptide 2 (SP2), and p6 (depicted in Figure 3). Proteolytic cleavage is orderly and sequential, with the initial site at the C-terminus of MA-CA-SP1, and the final cleavage event at the CA-SP1 junction [34,35,36,37]. Cleavage of Gag at each site is required for particle maturation and generation of infectious particles. A critical step in this process is the final cleavage event removing SP1 from CA, which serves as a conformational switch between the formation of the immature spherical shell and the mature, conical capsid [38].

Gag-Pol is the product of a −1 translational frameshift that produces a 160 kD Gag-Pol product [46]. The incorporation of Gag-Pol into the developing virion brings the critical enzymatic machinery into the particle, including PR, RT, and integrase (IN). Gag-Pol incorporation into the particle requires interaction with Gag, followed by co-trafficking of Gag and Gag-Pol to the site of assembly on the PM. Mutagenesis studies have indicated that CA-CA interactions are essential to Gag-Pol incorporation [47,48], although the RT region of Pol has also been found to contribute in some studies [49,50]. The ratio of Gag-Pol to Gag is important for normal assembly and maturation, as overexpression of Gag-Pol leads to premature intracellular activation of PR, reduced particle release, and reduced infectivity of released particles [51,52]. Some evidence points to more efficient incorporation of Gag-Pol into virions when Gag and Gag-Pol are synthesized in *cis* (from the same gag-pol RNA) [53]. Dimerization of PR domains in Gag-Pol during or immediately following the budding process leads to the ordered cleavage of Gag and Gag-Pol into their component subunits.

Purified HIV-1 Gag proteins alone at sufficient concentrations or combined with nucleic acids or inositol phosphates will self-assemble into either tubules, mature cores, or spherical particles [54,55,56]. This in vitro assembly property has facilitated structural studies of the mature and immature Gag lattice and has contributed to the identification and characterization of host factors that regulate assembly processes. Advances in structural data derived from NMR, cryo-electron tomography (cryo-ET), and protein crystallization studies have in recent years provided many new details of structural transitions occurring during maturation, all of which have provided new insights into functional domains present within the uncleaved Gag precursor and its cleavage products. Below we will discuss each individual domain within Gag, from N- to C-terminus, and use this organizational framework to highlight recent discoveries relevant to each domain. Refer to Figure 3 for a visual depiction of the domains and some of the structures discussed in this section.

## 5. MA: Regulator of Gag-Membrane Interactions and Env Incorporation

MA is the N-terminal segment of Gag and underlies the lipid envelope of the mature virus. The N-terminus of MA is myristylated, and this fatty acid modification is critical to the particle assembly process [57,58]. Mutations of the N-terminal glycine residue of MA prevent myristylation, leading to severe defects in assembly of HIV particles and eliminating viral replication in T cell lines [57,58]. Lack of myristylation leads to an altered subcellular distribution of Gag, shifting Gag from a prominent distribution on the PM to a more diffuse, cytoplasmic signal. The structure of MA was first solved by X-ray crystallography as a trimer with a N-terminal globular head, consisting of five α- helices, a 3_10_ helix, and a mixed β-sheet; followed by a long C-terminal tail that connects MA to the rest of Gag (Figure 3B) [59]. A highly basic region (HBR) consisting of conserved residues on the membrane-facing surface of the globular head of MA is an important contributor to membrane interactions [60,61,62,63]. Deletion or substitution of positively charged residues on this face disrupt PM interactions and can result in mistargeting of Gag to intracellular membranes [64]. NMR studies with myristylated MA demonstrated that the myristate group can adopt a sequestered or exposed conformation, and revealed that the exposed conformation was required for MA trimerization [65]. Inclusion of the CA domain in these studies (MA-CA) increased the exposed conformation of myristate. These structural studies supported the hypothesis derived earlier from studies of MA mutants in cell culture that proposed a “myristyl switch” mechanism to explain the enhanced membrane binding of full-length Gag compared to MA [63,66,67]. Together these and other studies led the model whereby Gag is a cytosolic monomer at low concentrations, with myristate sequestered. As Gag concentration increases in the cell, Gag multimerization occurs, triggering MA trimerization and myristate exposure that enhances membrane interactions. Both the HBR and myristate thus serve to anchor Gag to the lipid membrane.

The myristyl switch mechanism provided some insight into Gag–membrane interactions, but it did not explain why assembly occurs selectively on the PM of cells rather than on intracellular membranes. An important clue to this came from the work of Ono and Freed, showing that phosphatidylinositol (4,5) bisphosphate (PIP2) plays an important role in selective PM binding [68,69,70]. Depletion of PIP2 by overexpression of 5-phosphatase IV redirected Gag to intracellular membranes and reduced particle formation, while expression of a constitutively active form of Arf6 led to enhanced PIP2 on intracellular endosomes and also redirected Gag to these membranes [70]. Saad and coworkers subsequently provided the structural basis for MA interaction with PIP2 [61]. PIP2 was shown to bind directly to MA and adopt an “extended lipid” conformation, while also triggering the myristyl switch. This finding provided an explanation for Gag interaction with specific PM microdomains or membrane rafts, and supported a model in which PIP2 enriched on the cytosolic face of the PM provides a key site for the initiation of particle assembly.

Recent studies have shed light on additional mechanisms by which MA regulates the assembly process. MA was found to bind to cellular RNA in a manner that can block binding to liposomes that do not contain acidic phospholipids [71,72]. Cross-linking immunoprecipitation sequencing (CLIP-seq) studies demonstrated that MA binds to selective cellular tRNAs, and this interaction can inhibit the binding of Gag to membranes [73]. These data suggest that MA interactions with membranes may be inhibited by tRNA binding to residues within the HBR, blocking interactions with intracellular membranes and providing another potential explanation for selectivity of Gag binding to the PIP2-enriched PM. According to this model, both myristate sequestration and occlusion of the HBR of MA prevent premature interactions of Gag with intracellular membranes. When Gag concentration rises, multimerization results in myristate exposure and the HBR can interact with PIP2-enriched membranes, whose polar head groups displace tRNA from the HBR. Structural data supporting this model was provided by the co-crystallization of MA with tRNA, which revealed that residues in the HBR bind to the “elbow” or corner structure of the tRNA [74]. Mutation of single amino acid residues K32A or W36A abolished MA-tRNA interactions, but not PIP2 interactions, and resulted in redistribution of Gag from the cytosol to a more prominent PM distribution. These authors suggested that the prevention of premature membrane binding by tRNA allows more efficient viral replication, as mutant viruses unable to bind tRNA replicated with slightly reduced kinetics compared to wildtype [74].

The studies described above emphasize the role of MA in Gag-membrane interactions. Another essential role for MA is in facilitating the incorporation of Env. Deletions or substitutions within the globular head of MA eliminate the incorporation of Env with a full-length cytoplasmic tail (CT) into HIV-1 particles [75,76,77,78]. MA mutations that eliminate Env incorporation can be rescued by compensatory mutations within MA [79,80]. Prominent among the compensatory mutations is Q62R, a residue at the MA trimer interface. Because incorporation-defective mutants clustered around the perimeter of the MA lattice, it was proposed that steric hindrance of insertion of the gp41 tail into this gap is responsible for exclusion of Env [80]. According to this model, Q62R is able to alter the trimeric structure of the lattice and alleviate the steric clash of mutants such as 12LE with the gp41 CT. The importance of MA trimerization for Env incorporation was further supported when Tedbury and colleagues used a crosslinking strategy to demonstrate that MA trimers are present in HIV-1 virions, and that mutations that disrupted the trimer interface, including substitutions at T69 and L74, resulted in loss of trimerization and loss of Env incorporation [81]. A CT-truncated Env was still incorporated into virions lacking MA trimers, suggesting that accommodation of the CT by MA trimers is required for incorporation. These studies further supported the idea that in the absence of MA trimerization, Env is sterically excluded from the HIV-1 virion. The rescue of trimer-disrupting mutations by compensatory mutations restored viral replication, and at least for some compensatory mutations was associated with restored formation of the hexameric lattice of MA trimers [82]. A crystal structure of Q62R MA, however, revealed only very minor conformational differences from wildtype MA in the loops connecting helices I and II and helices V and VI [83]. New hydrogen bonding was seen at the trimer interface, indicating that Q62R stabilized trimer formation while not radically altering conformation. These studies together support a connection between MA trimerization and Env incorporation, while not yet providing conclusive evidence for a direct MA-CT interaction (discussed below in section on models of Env incorporation).

MA crystallizes on membrane monolayers into a hexameric lattice of trimers [84,85]. Recent work from the Briggs group allowed modeling of the structure of MA within immature and mature HIV-1 particles using MA lattice data derived by cryo-ET (Figure 3F) [43]. This work confirmed the presence of a hexameric lattice of MA in the immature virions and of large holes in the lattice at the six-fold axis, although the lattice was poorly ordered. Remarkably, in the mature virion MA also forms a hexameric lattice, but there are significant rearrangements. In the immature lattice, the HBR of MA faces the holes, while in the mature lattice a largely neutral surface surrounds the holes. The mature MA lattice is more ordered and is stabilized by PIP2 interactions [43]. The surprising discovery of a highly ordered MA lattice in the mature virion, and not just during the assembly process, suggests that there may be a post-entry function for MA, perhaps in some aspect of HIV replication following fusion and entry.

## 6. CA: Gag–Gag Interactions and Structural Transitions

CA is a defining structural component of the immature Gag lattice and of the conical capsid of the mature viral particle. Extensive structural information has been derived from X-ray crystallography, NMR spectroscopy, and cryo-ET detailing the interactions of CA in both the immature Gag lattice and in the mature capsid. The CA monomer is a 231 amino acid protein made up of two major domains: an N-terminal domain (CA_NTD_) and a C-terminal domain (CA_CTD_), connected by a small flexible linker (Figure 3C) [86,87,88]. The CA_NTD_ is made up of 7 α-helices and an exposed β-hairpin loop at the N-terminus that serves as the binding site for cyclophilin A [89]. In the context of uncleaved Gag, the N-terminal β-hairpin cannot form, and this region is disordered. The CA_CTD_ is required for capsid dimerization and Gag oligomerization and contains a 20-amino acid stretch called the major homology region (MHR) that is highly conserved in onco- and lentiviruses and in the yeast Ty-3 retrotransposon [31,90,91]. The crystal structure of the CA_CTD_ revealed an ovoid protein formed by an extended strand followed by 4 α-helices [86]. The MHR forms a strand-turn-helix motif that is packed against the C-terminal end of helix 2. Cryo-ET studies of immature particles reveal that Gag forms a curved hexameric lattice, with a radial distribution of Gag from the N-terminal MA region at the membrane to the C-terminus nearest to the center of the shell [92,93]. Surprisingly, the lattice was found to form an incomplete sphere, with areas of lipid bilayer lacking any underlying Gag protein [92]. Improvements in subtomogram averaging have more recently allowed for high-resolution structures of the immature lattice, showing well-ordered densities for CA and SP1 and allowing positioning of all α helices from the solved subunit structures [44]. Below the CA_CTD_ is a density representing SP1, which forms a 6-helix bundle in the immature capsid [94]. The central role of the CA-SP1 junction as a molecular switch controlling maturation of the capsid is discussed below. Upon cleavage of Gag by HIV-1 PR, a remarkable structural rearrangement occurs in CA and results in the formation of the mature capsid core, as the spherical immature capsid is transformed into the cone-shaped mature capsid. Comparisons of the structure of the mature vs. immature capsid lattice revealed striking differences, including a different relative orientation of the CA_CTD_ dimer and a dramatic difference within the CA_NTD_. Essentially all CA–CA interactions in the immature capsid shell appear to be broken during maturation, and the weight of evidence suggests that capsid disassembly into small oligomers and monomers of CA occurs, followed by capsid reassembly to create the mature capsid. Cleavage at both the N-terminus (MA-CA junction) and C-terminus (CA-SP1 junction) plays a role in core condensation. N-terminal cleavage allows formation of a β-hairpin structure that does not exist in the immature capsid and enhances mature capsid formation [95]. Several excellent reviews provide detailed discussions of the structural rearrangements in CA that occur upon maturation [96,97,98].

The mature CA lattice has a larger hexamer–hexamer spacing than the immature lattice [96,99,100], and the CA–CA interfaces that are present in immature particles are different from those present in the mature core. In the mature core, six CA_NTD_ monomers form an outer hexameric ring with a central 18-helix bundle that provides intra-hexamer stability, while the CA_CTD_ portions reside below the hexameric ring and contribute inter-hexamer interactions. Intrahexameric NTD–CTD interactions between adjacent CA molecules add further stability to the capsid [101,102,103]. The mature capsid core is somewhat pleomorphic in shape, modelled as a fullerene structure in which the hexameric lattice is closed through the incorporation of 12 CA pentameric vertices [104,105,106,107]. The major role of the mature core is to mediate post-entry steps of the replication cycle, encapsidating the viral genome together with NC and the viral enzymes RT and IN in order to deliver the preintegration complex to the nucleus for integration. The unique conical shape of the mature capsid may allow positioning of the incoming capsid on the nuclear pore complex (NPC) to facilitate nuclear entry, with the narrow end of the capsid inserting into the NPC prior to transport into the nucleus [108].

## 7. The CA-SP1 Junction: A Key Molecular Switch Regulating Assembly and Maturation

A role for SP1 in modulating HIV maturation was first revealed by mutagenesis, modeling, and NMR studies showing that the CA-SP1 junction forms an α-helical structure that is critical for particle formation [34,94,109,110,111]. Using an in vitro assembly model, Gross and colleagues showed that Gag polyproteins could form spherical particles representing the immature capsid or tubular and conical particles resembling the mature capsid, while upon deletion of SP1 only mature particle forms were observed [38]. Notably, in the ordered cleavage of Gag by HIV-1 PR, cleavage at the CA-SP1 site is the final proteolytic event, and blocking this cleavage through mutagenesis results in the formation of non-infectious particles and prevents mature capsid condensation [35]. These studies together established that cleavage of CA-SP1 acts as a molecular switch in triggering the capsid maturation process. Many additional insights into the critical role played by CA-SP1 have come from studies of a category of compounds known as maturation inhibitors (MIs). The small molecule 3-O-(3′,3′-dimethylsuccinyl)-betulinic acid, also known as PF-74 or bevirimat (BVM), was developed as an HIV inhibitor that acts by blocking virion maturation [112,113,114]. The molecular target of BVM is the CA-SP1 cleavage site, so that in the presence of the drug this final proteolytic cleavage event is prevented. BVM can bind to immature particles but not mature particles, consistent with a model in which its binding site is disrupted upon proteolytic cleavage of the CA-SP1 junction [115]. Viruses with resistance to BVM were shown to contain mutations at the CA-SP1 junction that disrupt interaction with the compound, indicating a direct binding between this region and BVM [116,117]. Thus, studies with BVM as well as other MIs highlight the critical role of the CA-SP1 region in regulation of HIV-1 assembly and maturation. Structural studies confirmed the presence of a six-helix bundle at the CA-SP1 region, revealing that MIs stabilize this structure and suggesting that MI resistance mutations act to destabilize CA-SP1 [40,118]. BVM acts to stabilize the six-helix bundle through direct binding to a central channel within the bundle [40,118,119]. Furthermore, a compensatory mutation for MI-dependent viral mutants, the SP1-T8I mutation, extends and stabilizes the six-helix bundle and allowed visualization of the bundle extending to the N-terminus of NC by cryo-ET [120,121]. Together, the studies with MIs have shown that the CA-SP1 region is a dynamic one, with an equilibrium between an unfolded or disordered structure and the rigid six-helix bundle. The disordered conformation allows HIV-1 PR to access the CA-SP1 cleavage site, whereas the rigid or MI-stabilized bundle does not allow access and prevents cleavage (reviewed in [122]). The critical nature of the CA-SP1 region in regulating the transition between immature and mature virions was further emphasized by the recent discovery of IP6 as a host molecule that regulates the assembly process (discussed in a subsequent section of this review).

## 8. NC: Assembly Effects and Genomic RNA Packaging

NC is a 55 amino acid basic protein characterized by the presence of two CCHC zinc finger motifs, separated by a short linker. The presence of zinc finger motifs in NC is a highly conserved feature of retroviral Gag proteins [123]. Each of the zinc finger motifs of HIV-1 NC coordinates a zinc ion at high affinity [124,125,126]. Surrounding the zinc fingers are small domains highly enriched in basic residues. NC has been shown to perform many important functions in the HIV-1 lifecycle, including acting as a nucleic acid chaperone, playing an essential role in reverse transcription and recombination, contributing to multimerization of Gag during the assembly process, and serving as the primary region of Gag responsible for packaging of the viral genomic RNA (reviewed in [127,128,129,130]). For the purpose of this review, we will focus on the contributions of NC to Gag–Gag multimerization and assembly, and its essential role in packaging of the genome.

Early observations of the importance of NC to assembly came from the study of Gag proteins bearing C-terminal deletions or site-directed mutations within NC. Deletions of most of NC led to particle assembly defects, including diminished particle formation, altered morphology, and altered sedimentation characteristics [131,132,133]. Deletions of NC of Rous sarcoma virus (RSV) produced light density particles, and the production of particles of normal density could be rescued by substituting portions of NC from heterologous Gag proteins including HIV-1 [134]. This property seemed most related to preservation of the highly basic residues, and not the zinc finger structure itself. In vitro assembly studies supported a role for NC in enhancing assembly of purified CA-NC (vs. CA alone) into cylinders [135]. These and other observations led to the concept of the “I” or interaction domain within NC, in which NC contributes an important component to Gag–Gag interactions [136,137]. An important insight supporting the role of NC in promoting Gag–Gag interactions came from the replacement of NC by leucine zipper domains. The assembly defect observed upon deletion of NC was rescued by replacement with a zipper domain, in the absence of RNA packaging [138,139], providing strong evidence that NC plays a role in assembly through facilitating Gag–Gag interactions. NC in the context of HIV-1 Pr55^Gag^ plays an apparent role in membrane interactions, as Gag containing this portion of NC was found more prominently in a PM distribution [140,141]. However, it is difficult to separate this effect from the contribution that NC makes to Gag–Gag multimerization and formation of new particles on the PM. These studies establish that there is an assembly function localized within NC, most likely representing its effect on promoting Gag–Gag interactions, with an indirect effect on subcellular distribution as both myristate exposure and Gag multimerization itself enhance membrane interactions.

While the ability to replace the assembly function of NC with a protein interaction domain suggested that the RNA binding function of NC was not absolutely required for assembly, recent evidence supports a model in which NC–RNA interactions nucleate assembly events in the cell [142]. NC interacts with gRNA through specific contacts with the psi or packaging signal contained in the 5′ untranslated region or UTR [13,19,143]. Two copies of the gRNA are packaged, through specific recognition of the RNA dimer by NC as already discussed in this review [144]. Gag can also package cellular RNAs and assemble virions in the absence of gRNA, prompting the question of whether the Gag–gRNA interaction in infected cells is an important nucleating event in assembly or is simply a requirement for infectious particle production without a clear role in assembly [145,146]. Using a reporter system that uncoupled RNA signal from assembling Gag or Gag-Pol signal in single virion analysis, Dilley and colleagues asked if RNA influenced the efficiency of particle production [142]. They found that the gRNA provided in trans enhanced production of particles, and that this RNA-dependent enhancement required the specific psi signal and interactions with HIV NC. The efficiency of this interaction allows the HIV-1 RNA, which represents a small fraction of total cellular RNA, to preferentially induce HIV assembly events, thus resulting in infectious virions with packaged gRNA. Another study titrated in psi-containing RNA or control RNA with Gag in vitro, and assessed their relative abilities to nucleate particle assembly [147]. In this in vitro assembly reaction, psi-containing RNA was able to more efficiently nucleate particle assembly, in agreement with the results from cell-based studies. More recently, cell-based complementation techniques were employed to define the contribution of Gag:Gag multimerization or PM binding domains to RNA packaging, using constructs that clearly separated multimerization from RNA binding. Remarkably, mutations that disrupted either PM binding or Gag:Gag multimerization (such as disrupting myristylation or removing the CA_CTD_) resulted in severe RNA packaging defects [148]. This work suggests that packaging occurs on or near the PM and requires the ability of Gag to multimerize.

Structural studies reveal that NC binds to individual stem loops within the psi region of the 5′ UTR, creating tight interactions between the zinc fingers and guanosines in the RNA [12,13]. As already discussed in the section on RNA dimerization and packaging in this review, the full-length unspliced HIV gRNA serves as both the packaged genome and as the template for translation of the Gag and Gag-Pol proteins. The basis for selection of RNA for either packaging vs. translation was proposed to be due to differences in base pairing within the DIS, presenting either a structure with the intact stem loops in which the initiation codon for Gag is at the base of SL4 and not ideally positioned for translational initiation, or an alternate structure where the DIS palindrome is occluded but the initiation codon is exposed [18,149]. Supporting evidence for the existence of the packaging structure was provided by NMR, showing how residues of the major splice donor and translation initiation are sequestered [150]. These developments in packaging of RNA including nucleation of assembly have been recently reviewed [151]. The problem of how gRNAs were selected for packaging or translation was then solved by the aforementioned finding that the transcriptional start site for gRNA determines the fate of the RNA through its influence on the structure of the capped leader RNA [28,29,30]. Thus, the structural basis of HIV-1 gRNA packaging versus translation has now been explained in a very compelling way.

## 9. P6: HIV’s Exit Strategy and Incorporation of Vpr

At the C-terminus of Gag is p6, a 52 amino acid protein that is best known for the presence of late or “L” domains and for providing the packaging determinants for the Vpr protein. The first indication that p6 played a role in particle budding or release came from the finding that deletion of p6 produced a defect in particle release, with particles remaining attached to the PM by a thin stalk [152]. The critical domain regulating release was mapped to a small PTAP motif by site-directed mutagenesis [153]. The Gag proteins of other retroviruses were also found to have short motifs contributing to virus release, including PTAP, PPXY, and YPXL motifs. The ability to contribute to very late events (particle release) in the virus lifecycle led to the term late domains for these motifs [154]. Subsequently it was shown by several groups that PTAP binds to the cellular protein TSG101, a component of the ESCRT-I complex, thus linking Gag to the ESCRT machinery [155,156,157]. The ESCRT machinery is a multi-subunit membrane remodeling complex that plays a key role in the biogenesis of intraluminal vesicles within the multivesicular body (MVB) [158,159]. A second L domain is also found in HIV-1 p6, a YPXL motif that binds to ALIX, which is a component of the ESCRT-III complex [160]. TSG101 plays a dominant role in mediating particle release, but recruitment of ALIX was found to also contribute to HIV-1 replication in T cells and macrophages [161]. Through its L domain interactions with ESCRT components, Gag co-opts the cellular machinery for membrane scission from its role in the late endosome/MVB to assist with viral particle release from the PM. A subset of ESCRT components is required for release of HIV-1 and other retroviruses, while other ESCRT components may play an accessory role. In particular, ESCRT-III components including CHMP2 and CHMP4 and the vacuolar protein sorting 4 (VPS4) ATPase are recruited to the site of particle budding, where they mediate the final membrane scission event separating the thin “neck” of the budding virion from the PM. Remarkably, the ESCRT-III components organize into spiral filaments on the inner membrane of the neck of budding particles, performing “scission from within” [162,163]. The recruitment of components of the ESCRT machinery for the purpose of promoting HIV-1 particle release has been summarized in multiple reviews [159,164,165].

The ubiquitylation of cellular proteins serves as a signal for recognition by the ESCRT machinery, prompting the natural hypothesis that ubiquitylation of Gag may play a role in late stages of the lifecycle. Supporting this, fusion of ubiquitin to equine infectious anemia virus (EIAV) Gag can compensate for the lack of a late domain, and fusion of a deubiquitinating enzyme to HIV-1 Gag leads to diminished particle release [166,167]. Although the role of Gag ubiquitylation remains uncertain, these studies suggest a potential role in ESCRT recruitment.

Recent work on the role of ESCRT in HIV-1 release has shown that ESCRT-I is not just a link to ESCRT-III, but plays an essential mechanical role. Flower and colleagues determined the crystal structure of a subset of ESCRT-I components (TSG101-VPS28-VPS37B-MVB12A), demonstrating a helical assembly [168]. The helical structure itself was required for HIV-1 release, suggesting a model in which ESCRT-1 provides a helical nucleating platform upon which downstream ESCRT complexes assemble. Another area of recent investigation relates to the dynamics of ESCRT recruitment, assembly, and disassembly. Using high-speed total internal reflection fluorescence (TIRF) imaging, Gupta and colleagues identified distinct stages including recruitment of ESCRT components followed by VPS4-mediated disassembly of the complex that was catalyzed by Vps4 and required ATP hydrolysis [169]. Disruption of ALIX binding altered the sequence of events, which is normally characterized by a single phase of recruitment, and instead led to a “stuttering” recruitment of multiple rounds of ESCRT complex to the same VLP [170]. The exact meaning of this is uncertain, but it implies that ALIX plays a role in the dynamic sequence of fission events that will require further investigation. Of note, HIV-1 Gag lacks a PPXY-type late domain, which links some retroviral Gag proteins to the NEDD4 family of ubiquitin ligases. NEDD4L has been shown to stimulate HIV-1 budding and rescue late budding defects, indicating that K63-linked polyubiquitin chains play some role in particle release [171,172]. Mercenne and colleagues identified angiomotin as a host adaptor protein that binds to NEDD4L and links it to Gag in developing particle buds [173]. Depletion of angiomotin led to an arrest of particle budding at a stage earlier than TSG101 depletion, leading to a half-moon appearance of buds on the membrane. This suggests that angiomotin serves as a NEDD4L adaptor acting at a stage prior to ESCRT recruitment and function, and further implicates ubiquitylation in the events leading to particle release. The structural basis for interaction of the angiomotin PPXY motifs with NEDD4L WW domains was recently reported [174].

The second major function attributed to p6 is the recruitment of Vpr into the virion particle [175,176]. The primary function of Vpr in the HIV lifecycle remains unclear at this time, despite numerous observed effects highlighted by its ability to assemble a CRL4^DCAF1^ E3 ubiquitin ligase complex and target multiple cellular proteins for degradation [177]. Deletion of p6 leads to loss of Vpr packaging, which normally occurs robustly, with 1 Vpr molecule packed per every 7 Gag molecules [178]. Multiple Vpr binding sites within p6 have been identified, including FRFG, ELY, and LXXLF motifs [179,180,181,182,183,184]. An alanine-scanning mutagenesis approach showed that the FRFG and LXXLF motifs both contributed to Vpr particle incorporation, and also showed that oligomerization of Vpr contributes to its packaging [185]. This study also suggested that p6 plays no role in early events of the HIV-1 lifecycle and is rapidly lost following cellular entry.

## 10. The Role of IP6 in Formation of the Immature Gag Lattice and the Mature Capsid

The highly regulated processes of formation of the immature Gag lattice, cleavage of Gag by the HIV-1 protease during and after budding, and reassembly of the mature capsid containing the viral genome and reverse transcription and integration machinery has been described above. A recent fundamental discovery in the field is the role of inositol hexakisphosphate (IP6) in formation of both the immature lattice and the mature capsid. IP6 is a small polyanion also known as phytic acid, which is present in all mammalian cells at concentrations of 10–40 μM [186]. Inositol phosphates including IP6 had been found to enhance the in vitro formation of immature virus-like particles, with data suggesting IP6 interactions with basic regions of Gag within MA and NC [54,187,188]. Many years after these studies, advances in cryo-ET revealed new clues into the mechanism by which IP6 enhances HIV-1 assembly. An unknown density was first seen within the central pore of the immature CA-SP1 lattice resolved by cryo-ET and subtomogram averaging, and this density was coordinated by 12 lysine residues arranged in two rings (six K290 residues and six K359 residues of the CA_CTD_ and CA_CTD_-SP1 helix, respectively) [40,44]. This density was subsequently identified by Dick and colleagues as IP6 using a variety of structural, biochemical, and cell-based approaches [189]. Remarkably, knockout of inositol pentakisphosphate 2-kinase (IPPK), the enzyme responsible for the last step of IP6 synthesis, in a mammalian (293FT) cell line resulted in markedly lower production of infectious particles. Using a microcrystal strategy, the CA_CTD_-SP1 protein in the presence of IP6 formed flat hexagonal crystals from which a crystal structure was derived. This showed one IP6 molecule per hexamer interface as suggested by the prior density seen in cryo-ET studies from immature virions, with each phosphate coordinated by the two lysine rings formed by K290 and K359. The IP6 form seen in the crystal was most compatible with myo-IP6, where five phosphates are in the equatorial plane (coordinated by K290 ring) and one in the axial position (coordinated by the K359 ring) (Figure 4A, left). These data support a model where IP6 stabilizes the six-helix bundle, promoting the formation of the immature Gag hexamer. This role for IP6 did not require either MA or NC, suggesting that IP6 stabilizes the immature Gag lattice and is a major determinant of HIV-1 assembly. IP6 was subsequently shown to act in a similar fashion to promote the formation of EIAV particles [190].

The role of IP6 in immature lattice formation was not the only major finding from this work. The investigators also examined the role of IP6 on mature capsid assembly [189]. As discussed above, upon cleavage by HIV-1 protease the interactions of the immature Gag lattice are disrupted, and CA–CA interactions reform to create the mature hexameric lattice. In this structure, a central pore is also formed that is ringed by positive charges, in this case contributed by R18 of CA. Addition of IP6 to CA in vitro promoted the formation of mature CA hexamers, and mutation of R18 to alanine prevented this effect. Crystal structures revealed IP6 within the central pore of the mature hexamer, either just above (favored) or below the R18 ring. This suggested a model in which IP6 stabilizes the 6-helix bundle through binding to K290 and K359 to enhance formation of the immature particle, and then promotes formation of the mature capsid through interactions with the R18 ring after cleavage. In both cases this polyanion serves to neutralize what would otherwise be repulsive positive charges located at the center of the hexameric ring. The remarkable transition of IP6 from its role in the immature Gag lattice to the mature lattice is depicted schematically in Figure 4A, with structures of IP6 within the mature lattice shown in Figure 4B. Notably, IP6 is concentrated within developing virions as the immature particle forms, providing sufficient molecules to efficiently facilitate mature capsid formation upon proteolysis of Gag. Subsequent studies have confirmed the essential role of IP6 in particle formation and production of infectious particles [191,192,193]. Binding of IP6 is adjacent to the binding site of BVM on the 6-helix bundle. When IP6 was added in the presence of BVM in an in vitro maturation experiment, an additive effect on inhibition of cleavage at the CA-SP1 site was observed, suggesting that IP6 and BVM are not competing for the same binding site [194]. An excellent review of the important role played by IP6 in retroviral assembly has recently been published by Obr and colleagues [195].

## 11. Gag Protein Trafficking to the PM

A specific role for retroviral MA in determining the site of assembly of particles was shown years ago by Rhee and Hunter [196]. They demonstrated that a single amino acid change (R55W) within M-PMV MA dramatically altered the site of particle assembly from an intracellular site (type D assembly) to assembly on the PM (type C assembly). HIV-1 Gag assembles on the PM in a manner that is similarly dependent on trafficking and membrane interactions mediated by MA. As discussed in the section on MA, both myristate and basic residues on the globular head of MA mediate interactions with membranes, and interaction with PIP2 plays a specific role in Gag binding to the inner leaflet of the PM. What is less clear is how Gag travels to the PM from its site of translation on cytosolic ribosomes. Visualization of assembly intermediates within the cytoplasm or on intracellular membranes has been reported in the past [197,198]. However, live cell imaging approaches using TIRF microscopy have generally not supported this model, and instead reveal formation of the Gag lattice starting on the PM and progressing through particle assembly and release [199,200,201]. The cytosolic blush of Gag-GFP seen upon live imaging suggests that Gag is diffusely present throughout the cytoplasm, and only upon nucleation of assembly at the PM, aided by gRNA, does particle assembly begin [142]. However, there is not uniform agreement that Gag monomers passively diffuse to the site of assembly on the PM. Gag oligomers that traffic toward the PM have been proposed in some studies [202]. A more provocative finding comes from the report of Gag trafficking to the nucleus, first reported for RSV Gag by the Parent laboratory [203,204]. For RSV Gag, nuclear localization has been linked to efficient packaging of gRNA, and it has been proposed that Gag is transiently imported into the nucleus where it collects gRNA, followed by nuclear export of the ribonucleoprotein complex and transit to the PM for assembly. These investigators have also reported the presence of HIV-1 Gag with viral gRNA in nuclei within HeLa cells, suggesting that a similar process of nuclear transit could occur for HIV-1 [205]. This finding would lead to a radically different model of Gag intracellular trafficking and gRNA packaging, and will require further evaluation and validation before it can be widely accepted.

We have focused on Gag trafficking to the PM for this review of HIV-1 assembly. An alternative site of assembly has been proposed for HIV in macrophages. Macrophages infected with HIV-1 in cell culture develop an intracellular structure with characteristics of both the MVB and the PM in which viral particles accumulate [206,207,208,209,210]. This apparent organelle, generally termed the virus-containing compartment or VCC, features thin tubules connecting to the PM [206,211]. The VCC has been proposed as an intracellular site of particle assembly and budding. Alternatively, the VCC may serve as a site of particle sequestration in macrophages after assembly at the PM. The prominent role of tetherin and Siglec-1 in VCC formation, along with the ability of Siglec-1 on the surface of macrophages to capture and internalize exogenous HIV-1 particles into the VCC, supports this view of the nature of the VCC [212,213].

## 12. Env Protein Trafficking and Particle Incorporation

An essential element for the formation of infectious particles is the incorporation of the HIV-1 Env protein. Env is initially synthesized on ER-associated ribosomes as a gp160 precursor polyprotein. Gp160 bears a signal peptide on its N-terminus that directs insertion into the ER lumen, and insertion concludes upon reaching the membrane anchor within the transmembrane (TM) portion of the protein [214,215]. Mutations within the signal peptide have been shown to alter Env incorporation, and in some cases to affect glycosylation and neutralization sensitivity [216,217]. N- and O-linked glycosylation and folding of the Env ectodomain occur within the ER. Env (gp160) monomers form trimers in the ER, and this trimerization facilitates transit through the secretory pathway [218,219,220]. Glycan processing with the addition of complex glycans takes place in the medial and late Golgi, while some glycans remain “underprocessed” and are present as high-mannose or oligomannose glycans [221]. Cleavage of gp160 to its component gp120 (SU) and gp41 (TM) subunits occurs within the Golgi and is mediated by a furin-like protease [222,223,224]. The Env glycoprotein that reaches the cell surface is therefore a trimer of heterodimers of gp41 and gp120 and is heavily glycosylated. Upon reaching the PM, Env is rapidly endocytosed [225,226]. Endocytosis requires distinct motifs in the cytoplasmic tail (CT), including a membrane-proximal Yxxφ motif (x = any amino acid, φ = amino acids with bulky hydrophobic side chains, such as leucine, phenylalanine, valine) and a C-terminal dileucine motif, both interacting with clathrin/AP-2-dependent endocytosis machinery. The rapid endocytosis of Env seems counterintuitive as a viral replication strategy, especially for a virus that assembles its infectious virions on the PM. It has been postulated that endocytosis is a form of immune evasion, limiting Env presentation on the cell surface and potentially on virions. However, it may also represent a unique trafficking strategy for Env that allows Env and Gag to co-target to a common site of assembly, discussed further below.

As the principal neutralizing determinant of the virus, Env has undergone intensive investigation, including detailed structural studies of purified Env proteins in the form of monomeric gp120, native Env trimers, and Env trimers in complex with receptor CD4 and an increasing variety of broadly neutralizing antibodies (bNAbs) [227,228,229,230,231]. For the purpose of understanding cellular trafficking of Env, however, it is necessary to focus on TM and in particular on the CT. Gp41/TM includes the fusion peptide, an extracellular domain (ectodomain) that mediates gp120 interactions, the membrane-proximal external region (MPER), a transmembrane domain, and the long C-terminal CT. Lentiviruses including HIV-1 encode extremely long CTs, which has important implications for intracellular trafficking and for the physical accommodation of the tail by the Gag lattice [232,233]. The HIV-1 Env CT is 150 residues and has been modeled from multiple studies to include an unstructured region followed by three α-helical segments known as lentiviral lytic peptides (LLPs). For historical reasons these LLPs are numbered somewhat counterintuitively from N to C-terminus as LLP2, LLP3, and LLP1 [234]. The structure of the CT was first solved by the Saad laboratory, using a micellar solution and NMR spectroscopy [235]. The N-terminal region of the CT did not show a regular structure and was not associated with membrane, while the C-terminal 105 amino acid stretch formed three consecutive amphipathic helices that were tightly associated with membrane. The overall organization derived from this structure revealed an N-terminal loop exposed to the cytosol, while the LLP2, LLP3, and LLP1 helices are largely extended and buried within the membrane, with some exposed basic regions. A somewhat different structure was solved by the Chou group, using a construct that included the TM and CT, where the α-helical segments formed two concentric rings within the membrane bilayer forming a structure referred to as the CT baseplate [236]. Physical interactions occur between the C-terminal portion of TM and the amphipathic LLP2 helices that form the inner ring, and this was proposed to explain how the CT can influence the structure of the ectodomain and alter sensitivity to bNAbs.

### Models to Explain Env Incorporation

Four potential models to explain how Env trimers can be incorporated into developing particles have been proposed by the Freed group [232]. In the *passive incorporation model*, Env is present on the PM, and is passively acquired as the Gag lattice forms, with no specificity. The fact that truncations of the CT still allow Env incorporation into developing particles for infectious virion formation in some cell types supports a passive model of Env acquisition [75,76,237,238]. The *direct Gag-Env interaction model* postulates that a direct interaction between MA and the CT mediates Env particle incorporation. This model would potentially explain why mutations or deletions within MA lead to a loss of Env incorporation [75,76,78,239], and why truncation of the CT rescues MA mutants that cannot incorporate full-length Env [75,76,240]. This model posits that MA directly interacts with the full-length CT to direct incorporation, while a truncated CT can rescue MA mutants and does not require direct MA binding. A direct biochemical interaction between MA and Env was reported by one laboratory, although little additional biochemical evidence has been forthcoming in support of this model [241]. A more recent study utilizing MACA proteins and MA trimerization mutants, however, has provided additional support for direct binding between MA and CT [242]. Additionally, in support of the direct interaction model is the finding that maturation is required in order for Env to mediate viral fusion, suggesting that Gag constrains the Env ectodomain through contacts with the CT. Truncation of the CT relieves the constraint on fusion mediated by immature particles, apparently by removing the Gag–CT interaction [243,244,245]. The *Gag-Env co-targeting model* explains Env incorporation through trafficking of Env to a specific microdomain on the PM for interaction with Gag. Supporting this model is the strong data indicating that there are distinct cell type-specific differences in Env incorporation. In some transformed epithelial and T cell lines, such as 293T, HeLa, COS, and MT-4, CT-truncated Env is readily incorporated into developing particles. In contrast, other T cell lines including Jurkat, H9, and CEM along with primary T cells and macrophages all require an intact CT for incorporation and to support viral replication [237]. Cell types that allow incorporation of truncated CT Env are referred to as “permissive”, while the primary cell phenotype is “restrictive”, not supporting the incorporation of Env with a truncated CT [232]. The reason underlying a permissive or restrictive phenotype remains unclear, although the implication is that specific cellular factors regulating Env trafficking may differ between cell types. The potential of specific microdomains to serve as HIV-1 assembly sites is supported by the existence of lipid rafts, domains that are enriched in cholesterol and saturated fatty acids, on the PM. Potentially Gag and Env could co-target to these microdomains for coordination of assembly. Gag is found in detergent-resistant membrane fractions (DRMs) upon membrane fractionation, a characteristic used to define lipid raft components [246,247,248]. However, Gag does not segregate on flotation gradients classically as a lipid raft component [249], and although evidence for Env association with rafts has been presented, it is not particularly strong [250,251]. The most convincing argument that lipid rafts are preferentially organized and incorporated into budding particles comes from studies of the lipid composition of virions, which show enrichment of raft components cholesterol, glycosphingolipids, and phosphoinositides within the viral membrane [252,253]. Potentially, Gag and Env co-targeting to a particular subset of lipid rafts could explain specific incorporation of Env into HIV-1 particles. The last possible model as outlined by the Freed laboratory is the *indirect Gag-Env interaction model*, in which a host adaptor protein binds both to the CT and to MA. While proteins that interact with the CT have been identified, there is no clear evidence for a protein adaptor that directly links Gag and Env. Some of the CT interactors have been implicated in trafficking of Env, and may be more relevant to the co-targeting model of Env incorporation as discussed below.

Recent studies have shown that MA trimers are present in virions, and that MA trimerization plays an important role in Env incorporation as discussed in the MA section above [81]. Tedbury and colleagues identified MA trimers, and created mutants designed to disrupt the MA trimer interface. Viruses deficient in MA trimerization were deficient in incorporation of Env bearing an intact cytoplasmic tail, while particle incorporation and infectivity could be rescued by CT truncation. This finding supports a model where the long CT is accommodated by the MA trimer lattice but sterically excluded in trimer-defective particles. Further work in this area generated revertants within MA that restored trimer formation and Env incorporation [82]. Notably, some compensatory mutations were within the trimer interface, while others were distant from the trimer interface. These findings also support a model in which the central aperture in the MA hexameric lattice must accommodate the long CT, and potentially supports direct MA–CT interactions, while some MA mutations create a steric block to the CT that excludes Env. Interestingly, single molecule, super-resolution studies have shown that Env trimers accumulate toward the necks of budding virions [254,255]. This distribution required the long CT and an intact MA domain, indicating that lattice trapping of Env occurs on the neck that could support either a direct interaction or steric confinement model.

## 13. Host Factors Involved in Env Trafficking

Multiple host proteins have been implicated in Env trafficking in the cell. As already discussed, the AP-2 adaptor protein complex interacts with the membrane-proximal YXXφ motif in the CT and mediates endocytosis of Env from the PM [226,256,257,258]. Mutation of the YXXφ motif diminished endocytosis and led to increased cell surface levels of Env. Notably, the μ1 and μ3A subunits of the related AP-1 and AP-3 complexes can also bind to tyrosine-based signals in the Env tail, although with lower affinity [226], raising the possibility that multiple adaptor protein complexes may regulate Env subcellular distribution. The AP-1 adaptor complex has also been shown to interact with the CT through binding to a different portion of the tail, the C-terminal dileucine motif [259]. Mutation of this motif did not affect endocytosis rates, but instead led to diminished cell surface Env and a redistribution away from the cellular periphery. Because AP-1 is involved in sorting from either TGN or recycling endosomes, this suggested that AP-1 plays a role in outward movement of Env from one of these compartments to the PM [260]. However, this same C-terminal dileucine was also shown to interact with AP-2 and to contribute to clathrin and AP-2-dependent endocytosis of Env [261].

Interactions of the CT with multiple additional components of the intracellular trafficking pathways have been identified. Retromer is a host sorting complex that is engaged in retrograde transport of some cargo from endosomes to the Golgi, and has also been implicated in transport of some cargo proteins to the PM [262,263,264]. A direct interaction between the CT and the Vps35 and Vps26 components of retromer was identified by coimmunoprecipitation and in vitro binding assays [265]. The interaction was mapped to determinants within the C-terminal 100 amino acids of CT. Depletion of retromer led to an increase in cell surface Env and increased Env incorporation into particles [265]. This study suggests that retromer normally transports Env from endosomes to Golgi, while the increased appearance of Env on the PM following retromer depletion suggests that Env can recycle to the PM via another pathway that leads to productive incorporation into particles. Rab11-FIP1C (FIP1C) (also known as Rab coupling protein) is a member of a family of Rab11-interacting proteins that link cellular cargo to transport vesicles bound for distinct cellular locations [266,267,268,269]. FIP1C is implicated in recycling of cellular cargo from the endosomal recycling compartment (ERC) to the PM, including α5β1 integrin and EGFR [270]. FIP1C plays a role in HIV-1 envelope trafficking and particle incorporation [271]. Depletion of FIP1C led to defects in particle incorporation that were dependent upon an intact CT. Env with an intact CT redistributed FIP1C out of the ERC, suggesting formation of a transport complex, and the FIP1C-binding partner Rab14 was shown to be essential for Env particle incorporation. A subsequent study utilized FIP1C redistribution to map determinants within the CT and identified a tyrosine-based motif (YW_795_) that was critical for this effect and that resulted in a defect in Env incorporation and replication in restrictive T cell lines such as H9 [272]. The defect in incorporation mirrored that seen with near-complete CT deletion such as that seen with CT_del_144, suggesting that this motif is essential for cell type-dependent incorporation of Env. After prolonged culture in restrictive cells, a revertant virus bearing a single amino acid change within the C-terminal portion of CT (L_850_S) arose. This C-terminal mutation rescues the YW_795_SL loss-of-incorporation mutation, and restored replication in restrictive cells. The importance of Env trafficking through the ERC was further demonstrated by the ability of FIP1C C-terminal fragments to prevent Env incorporation in a dominant-negative fashion, arresting Env within the condensed ERC [273]. Together, these findings support a role for cellular factors, through a direct or perhaps indirect role of FIP1C, to mediate Env trafficking and particle incorporation. A model for Env incorporation proposed based upon the role of host trafficking molecules is shown in Figure 5. We note, however, that the sequential trafficking of Env remains under investigation, and the role of additional cellular factors in determining cell type-specific incorporation of Env seems likely.

## 14. Host Restriction Factors Active during HIV-1 Particle Assembly or Release

Host restriction factors are molecules that have evolved to limit or prevent viral replication, and constitute an innate host defense against viral pathogens [274]. Classically a restriction factor shows evidence of positive selection over evolutionary history, and most but not all are the products of interferon-stimulated genes (ISGs) [275]. Restriction factors have evolved to target many distinct steps within the viral lifecycle. In turn, viruses may escape from host restriction by altering their genetic makeup, including acquiring new “accessory protein” functions that counteract the restriction. Here, we will discuss restriction factors that act to limit HIV-1 replication during assembly, budding, or release of viral particles. Understanding the mechanism of action of restriction can provide insights into key features of this half of the lifecycle, including areas where the virus may be susceptible to new therapeutic interventions.

### 14.1. GBP5

A genetic screen for human proteins that met criteria for HIV restriction factors (under positive selection pressure, upregulated by interferons, and inhibit HIV replication) identified guanylate binding protein 5 (GBP5) as a new candidate restriction factor [276]. Exogenous expression of GBP5 significantly reduced the yield of infectious HIV-1, while not inhibiting particle production or release [277]. The effects of GBP5 are mediated by alterations in Env protein glycosylation and particle incorporation. GBP5 expression results in increased particle incorporation of uncleaved gp160 and diminished amounts of gp120. GBP5 appeared to have major effects on the infectivity of particles released from human macrophages. Surprisingly, counteraction of the restriction to replication induced by GBP5 could be overcome by inhibiting translation of Vpu through introduction of mutations in the *vpu* gene, resulting in enhanced translation of Env. The investigators postulated in this work that naturally occurring HIV isolates lacking an intact *vpu* gene may arise by selection to overcome restriction by GBP5, despite the benefits to the virus of having an intact Vpu to overcome tetherin-mediated restriction. The antiviral effects of GBP5 were shown to be mediated through inhibition of the cellular proprotein convertase furin, so this activity is relevant to other viruses whose glycoprotein processing depends upon furin cleavage [278].

### 14.2. MARCH8

MARCH8 is a member of the MARCH family of RING-finger E3 ubiquitin ligases. MARCH8 has been reported to downregulate the expression of numerous transmembrane proteins, including CD44, MHC-II, CD81, CD86, CD98, IL-1 receptor accessory protein, TRAIL receptor 1, Bap31, and transferrin receptor [279,280,281,282,283,284,285]. A genome-wide screen initially identified MARCH8 as a potential HIV-1 restriction factor [286]. Three MARCH proteins (MARCH1, 2, and 8) were subsequently linked to restriction through effects on envelope glycoproteins, including HIV-1 Env and VSV-G [287,288,289]. Similar to GBP5, overexpression of MARCH8 lead to diminished infectivity and reduced particle incorporation of gp120, while having no effect on p24 release [287]. Silencing of endogenous MARCH8 expression in macrophages led to a substantial increase in particle infectivity. While the mechanism was not entirely clear, the authors of this report postulated that MARCH8 interacts with Env and redirects it to an intracellular location, reducing Env at the PM and subsequent particle incorporation. A subsequent report from this group identified the site of Env retention as the Golgi, and defined a YXXϕ motif in the C-terminal domain of MARCH8 as required for its effects on HIV-1 Env [290]. The HIV-1 Env CT was not required for MARCH8-mediated restriction. This report also demonstrated ubiquitination of VSV-G by MARCH8. Additional insights into the mechanism of action of MARCH8 were provided by the Freed laboratory [291]. These investigators found similarly that the restriction on VSV-G-pseudotyped virions required its CT, while for HIV-1 Env (and both Ebola GP and SARS-CoV-2 S) the CT was not involved in MARCH8-mediated restriction. Restriction required the ubiquitin ligase activity of MARCH8, suggesting that direct ubiquitination of the CT leads to restriction of VSV-G, while effects on HIV-1 Env may occur through ubiquitination and degradation of a cellular trafficking factor. Altogether, studies of MARCH8 identify a potentially important restriction that acts at the level of Env trafficking and degradation and may be particularly important in myeloid cells.

### 14.3. PSGL-1

P-selectin Glycoprotein Ligand 1 (PSGL-1) is a dimeric, mucin-like glycoprotein that binds to P-, E-, and L-selectins and plays a role in leukocyte rolling on endothelial surfaces prior to transmigration. PSGL-1 is recruited to sites of HIV-1 particle assembly [292]. Liu and colleagues identified PSGL-1 as an IFN-γ-induced restriction factor in CD4+ T cells using a genome-wide proteomic screen [293]. They found that PSGL-1 inhibited reverse transcription in target cells and diminished the infectivity of released virions. The inhibitory effect was antagonized by the Vpu protein through the ubiquitination and proteosomal degradation of PSGL-1. This group subsequently reported that PSGL-1 acts to restrict actin dynamics and to arrest Env at the plasma membrane, resulting in virions with poor Env incorporation and reduced infectivity [294]. The inhibitory effect of PSGL-1 on HIV-1 replication was confirmed by Fu and colleagues, who documented a block at the stage of viral attachment to target cells; however, they did not observe an effect on reverse transcription [295]. PSGL-1 was readily incorporated into HIV-1 virions, and inhibited Env processing and incorporation. The negative effect of PSGL-1 on particle infectivity was not solely due to defective Env incorporation, but was mediated by a direct inhibition of virion attachment to target cells. PSGL-1 contains a large, extended ectodomain that is predicted to block viral glycoprotein interaction with cell surface receptors. Consistent with this idea, incorporation of PSGL-1 also inhibited the infectivity of murine leukemia virus and influenza virus. A similar effect was observed with a related mucin-like glycoprotein, CD43. PSGL-1 inhibited incorporation of SARS-CoV-2 spike glycoprotein in a similar way, preventing attachment of this virus to target cells, supporting the notion that this glycoprotein restricts the infectivity of a broad array of enveloped viruses [296].

### 14.4. IQGAP1

IQGAPs are a family of large, highly conserved scaffold proteins that regulate both microtubules and actin and play a role in many diverse cellular processes [297,298,299,300]. IQGAP1 has been shown to contribute to the budding of enveloped viruses including Ebola, Marburg, and Swine fever viruses, thus playing a positive role in viral replication [301,302,303]. The Goff laboratory found that IQGAP1 similarly plays a role in spreading infection of Moloney murine leukemia virus [304]. In contrast, this laboratory reported that IQGAP1 acts as an inhibitory factor for HIV-1 replication [305]. Overexpression inhibited HIV-1 budding, in the context of either expression of full-length provirus or of a Rev-independent Gag-Pol construct, while depletion of IQGAP1 enhanced HIV-1 particle release. IQGAP1 directly interacted with both the NC and p6 regions of Gag, and this interaction was required for its ability to inhibit release. An intriguing observation was that IQGAP1 altered the subcellular distribution of Gag, reducing the amount of Gag observed on the PM. The authors proposed that IQGAP1 may alter Gag trafficking, preventing Gag from reaching the PM, or may even actively remove it from the membrane to other locations in the cell.

### 14.5. Tetherin

Vpu is an accessory gene product of HIV-1 that was known to enhance particle output in many cell types, and to downregulate CD4, prior to discovery of its major host gene target tetherin. The inhibitory effect on particle release seen in Vpu-responsive human cells was shown to be a dominant restriction, suggesting that Vpu targeted a restriction factor limiting release [306]. This factor was identified as tetherin/BST2 by the Bieniasz and Guatelli laboratories [307,308]. Tetherin expression is highly upregulated by interferon-α expression [307,308,309]. Tetherin is an unusual type II membrane protein with a short cytoplasmic domain, a transmembrane domain, a coiled-coil extracellular domain, and a C-terminal GPI anchor [310]. The tetherin GPI anchor confers localization to lipid raft microdomains, and the molecule can link together rafts and the underlying actin cytoskeleton [311]. Tetherin acts as a direct physical tether for virions, linking the viral envelope to the PM and preventing the final “escape” of viruses from the producer cell [312,313,314]. Tetherin thus severely limits the spread of cell-free virus, and also reduces to a lesser degree cell–cell spread of HIV-1, at least according to several reports [212,315,316]. Vpu inhibits the action of tetherin by direct interactions between their transmembrane domains, resulting in intracellular sequestration and loss of tetherin on the PM [317]. *Vpu* genes are not present in HIV-2 or in SIV strains, except for the immediate simian precursor of HIV-1, SIVcpz. Countering of tetherin by these viruses instead occurs through the action of the SIV Nef protein, and the adaptations of SIV strains to overcome tetherin-mediated restriction have been implicated as major factors in simian lentivirus evolution, with the acquisition of Vpu contributing to the development of the HIV pandemic [318]. Tetherin-mediated restriction of particle release also acts as an innate sensing mechanism by the host, leading to signaling through NF-κB and upregulation of the host inflammatory response [319,320,321].

The restriction factors discussed above represent those that exert their effects during the assembly, budding, or release process. The sites of action of the restriction factors discussed here are presented schematically in Figure 6. It is important to note that other important restriction factors acting at distinct stages of the HIV-1 lifecycle must interact with viral components during assembly in order to be packaged into virions, and without this interaction they cannot exert their inhibitory effects on the virus. Prominent examples include APOBEC3G and 3F, which must be packaged through interactions with NC and the viral gRNA, and SERINC3 and 5, which are incorporated on the viral lipid envelope during the assembly process [322,323,324,325].

## 15. Summary and Outlook for the Future

Our understanding of the complexities of HIV-1 assembly continues to evolve, providing exciting new findings at every turn. Structural studies of mature and immature virions and intermediate structures have provided not only insights into pathogenesis, but also elucidate targets for the development of inhibitors of assembly. In recent years, major discoveries have been made through the identification and characterization of host factors acting during assembly, perhaps best exemplified by the critical role of IP6 in assembly and maturation. Although the field of HIV-1 assembly is relatively mature, there are many unanswered questions remaining. How exactly does Gag move from the site of translation to PIP2-enriched microdomains on the PM? What host factors regulate Env trafficking to determine cell type-specific incorporation of full-length Env? What additional host cell molecules regulate assembly events and remain to be discovered? It is safe to assume that many future discoveries await, and that they will continue to enlighten our understanding of the dynamic HIV-1 assembly process.

## Figures and Tables

**Figure 1 viruses-14-00478-f001:**
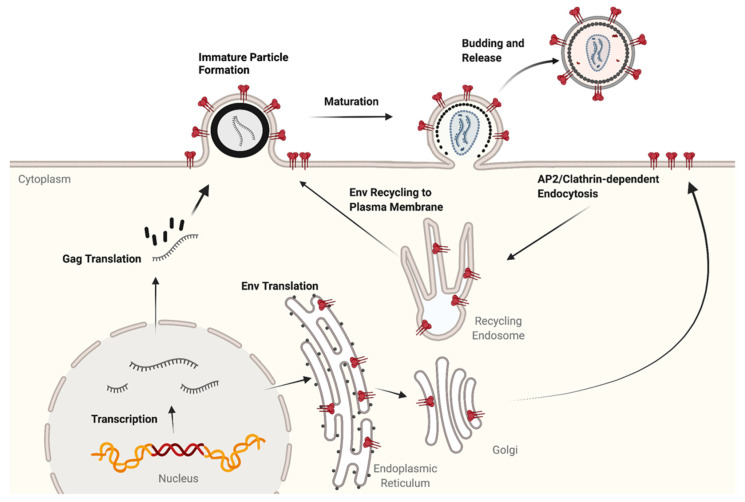
Schematic overview of HIV-1 assembly process. Gag and Env traffic via different routes to reach a common site of assembly on the PM. The path taken by Gag to reach the PM remains uncertain. Env travels through the secretory pathway to the PM, and then is rapidly endocytosed. Depicted here is a role for the recycling endosome in the outward movement of Env to the site of particle assembly. Following the assembly of the immature virion and packaging of the viral gRNA, cleavage of Gag occurs, leading to dramatic structural rearrangements and the formation of the mature virion. Adapted from HIV Replication Cycle by BioRender.com (2022).

**Figure 2 viruses-14-00478-f002:**
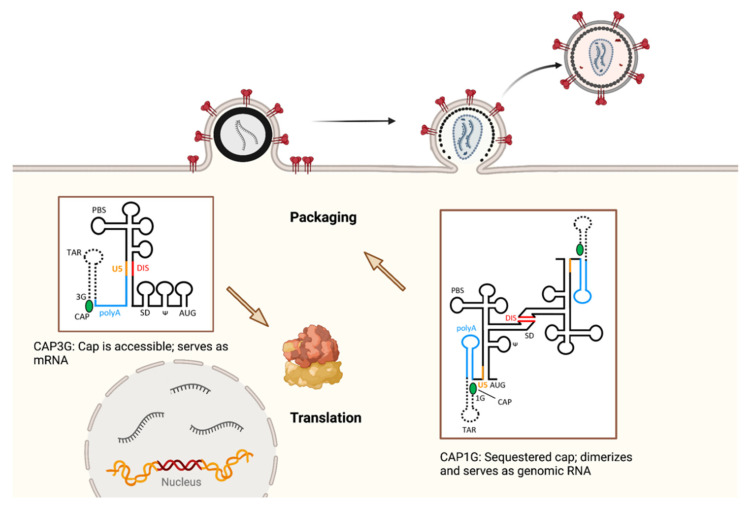
Selection of gRNA for packaging versus use as mRNA for translation. Note that in the CAP3G RNA, the 5′ cap is accessible and the RNA is monomeric, while in the CAP1G form the 5′ cap is sequestered and RNA dimerizes for packaging. RNA structure schematics modeled from [28]. DIS = dimer initiation site; SD = major splice donor; PBS = primer binding site. Figure adapted from HIV Replication Cycle by BioRender.com (2022).

**Figure 3 viruses-14-00478-f003:**
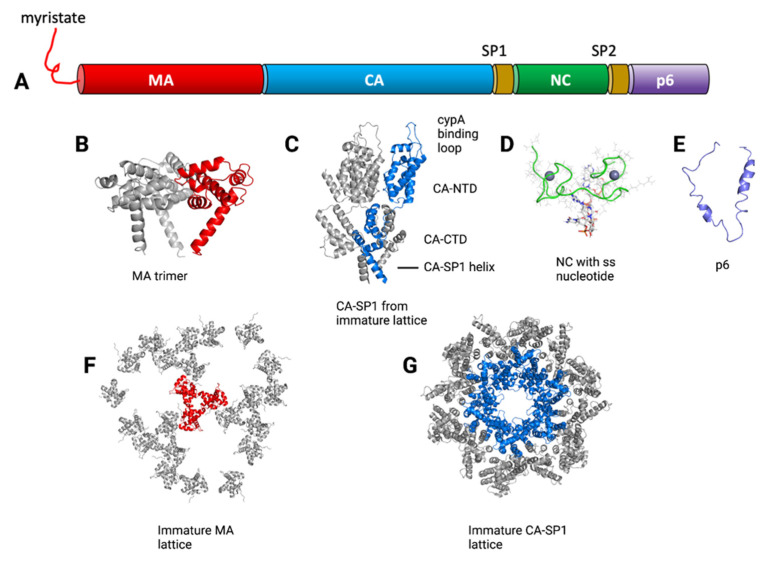
Gag polyprotein with cleavage products. (**A**) Schematic diagram showing cleavage products/domains in different colors. (**B**) MA trimer structure from PDB 1HIW [39]. (**C**) CA-SP1 structure, side view, with one CA_NTD_-CA_CTD_-SP1 colored in blue, from PDB 5L93 [40]. Note cyclophilin A (cypA) binding loop, CA_NTD_ and CA_CTD_ connected by flexible loop, and CA-SP1 helix. (**D**) NC structure with single-stranded oligonucleotide (ss nucleotide), from PDB 1BJ6 [41]. Zinc ions shown as spheres. (**E**) p6 structure from PDB 2C55 [42]. (**F**) Immature MA lattice, top view, with single trimer in red; from PDB 7OVQ [43]. (**G**) Immature CA-SP1 lattice, top view, with single hexamer colored blue, from PDB 4USN [44]. Structures rendered by the PyMOL Molecular Graphics System, Shrodinger, LLC. The arrangement of this figure was adapted from [45]. Figure compiled using BioRender.com (2022).

**Figure 4 viruses-14-00478-f004:**
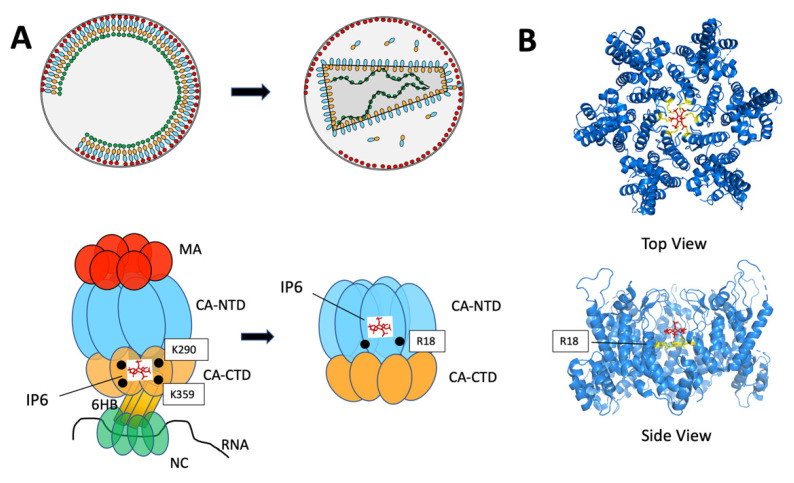
Role of IP6 in immature Gag lattice and in mature capsid core formation. (**A**) Schematic depiction of maturation of immature lattice to mature core formation, with structural model below showing position of IP6. Note that IP6 is coordinated by K359 and K290 rings in immature hexamer (**left**), while it interacts with R18 in the mature viral core. Figure adapted from findings of [189]. (**B**) Top and side views of IP6 in mature lattice, with positions of R18 residues indicated in yellow. IP6 is depicted in red. From PDB 6BHT [189]. Structures rendered by the PyMOL Molecular Graphics System, Shrodinger, LLC.

**Figure 5 viruses-14-00478-f005:**
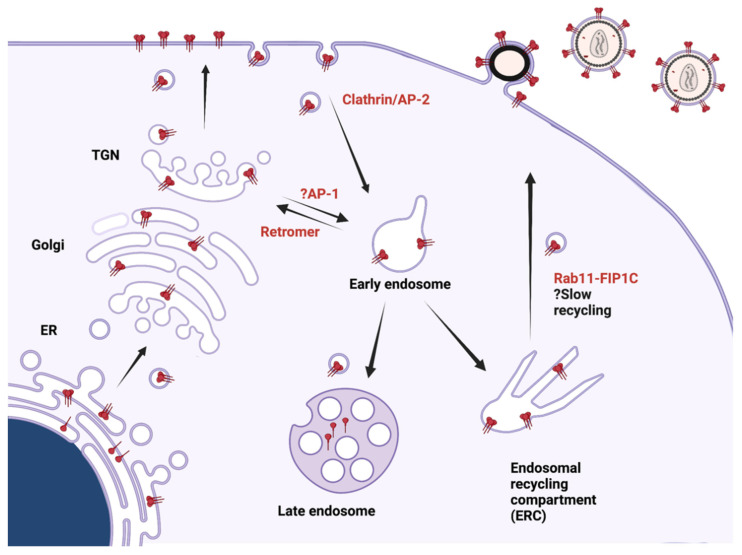
Host pathways implicated in Env trafficking and particle incorporation. Env is synthesized on ER-associated ribosomes, forming trimers in this organelle. As gp160 timers progress through the Golgi, they are glycosylated, and furin-mediated cleavage occurs in the TGN. The cleaved trimers reach the PM, followed rapidly by endocytosis into early endosomes. From here, multiple pathways are possible, including degradation in the late endosome/lysosome, retrograde transport to the TGN, and transport to the ERC. Evidence favors a role for outward recycling from the ERC in mediating Env incorporation into particles in a CT-dependent manner. Shown are roles for clathrin/AP-2, retromer, a potential role for AP-1, and outward recycling mediated by Rab11-FIP1C. Figure compiled using BioRender.com (2022).

**Figure 6 viruses-14-00478-f006:**
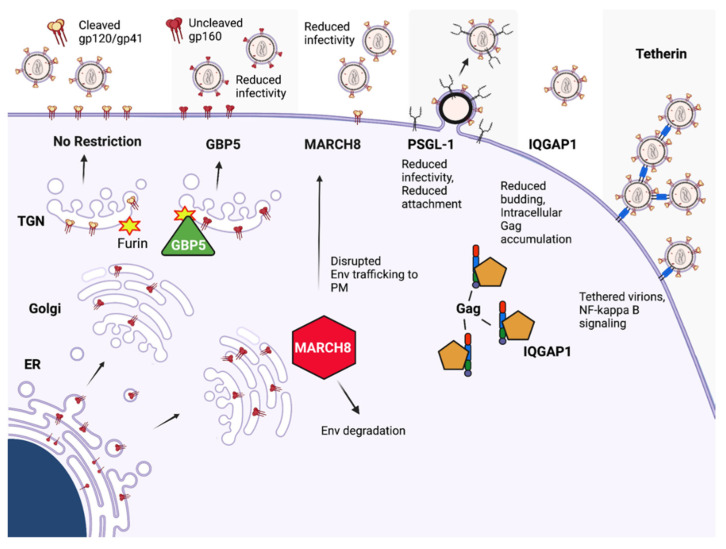
Restriction factors acting along the HIV-1 assembly pathway. On the left is pictured the unrestricted assembly of particles with fully cleaved Env. GBP5 and MARCH8 act to reduce particle infectivity through effects on Env, with a reduction in furin-mediated cleavage (GBP5) or disruption of Env trafficking and incorporation (MARCH8). PSGL-1 reduces Env incorporation but also reduces attachment to target cells. IQGAP1 binds to Gag and modulates particle formation, with reduced production of particles upon overexpression. Tetherin acts to prevent the release of fully formed particles, forming a physical tether to the PM and also acting as an innate immune sensor to stimulate signaling leading to NF-κB activation.

## Data Availability

Not applicable.

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
