# Peer review of "Advances in HIV-1 Assembly"

_viruses, 2022, doi:10.3390/v14030478_

Round 1

Reviewer 1 Report

In this review, Lerner et al. summarize literature of the HIV-1 assembly step and provide an up-to-date dose of knowledge on the various factors contributing to HIV assembly. The article is comprehensive and well-written and well-suited for the special issue: “Molecular Determinants of Enveloped Virus Assembly”. I wish if the authors consider the following points in the revised version.

  1. I suggest that the authors add more figures to illustrate the role for the viral and cellular proteins.
  2. Figure 3 panel D: text below the figure: NC with ss “oligonucleotide”
  3. Line 270: Perhaps it is worth describing the impact of the 62QR mutation on the structure of the MA trimer as shown the x-ray structure of 62QR MA mutant (Pubmed 33485964).
  4. Line 285: The cryo-ET study did not determine the structure of the MA within the lattice. Data provided low resolution models. It is more accurate to say models were reconstructed and MA structures were fit in the EM density.
  5. The manuscript will benefit from another round of editing to correct a few grammar issues. I am only providing some examples below:

Line 169: “Schematic colored” is not correct grammar.

Line 176: “Domain figure design” not clear.

Line 193: “will readily”

Line 259: “These authors”. Which authors?

Line 366: I would replace “destroyed” with disrupted.

Line 837: GBP5 instead of GBP.  

Author Response

We thank the reviewer for comments and suggested corrections. I address each comment below.

I suggest that the authors add more figures to illustrate the role for the viral and cellular proteins.

We added a figure (Figure 6) outlining the site of action of the restriction factors that act along the assembly pathway.

Figure 3 panel D: text below the figure: NC with ss “oligonucleotide”

Label added as suggested.

Line 270: Perhaps it is worth describing the impact of the 62QR mutation on the structure of the MA trimer as shown the x-ray structure of 62QR MA mutant (Pubmed 33485964).

We added a description of the crystal structure that is presented in this manuscript.

Line 285: The cryo-ET study did not determine the structure of the MA within the lattice. Data provided low resolution models. It is more accurate to say models were reconstructed and MA structures were fit in the EM density.

This is an important point. We revised the description to include the point about modeling the structure.

The manuscript will benefit from another round of editing to correct a few grammar issues. I am only providing some examples below:

Line 169: “Schematic colored” is not correct grammar.

We changed this line to be more clear.

Line 176: “Domain figure design” not clear.

We changed this line to more clearly represent the meaning.

Line 193: “will readily”

Corrected, dropped "readily"

Line 259: “These authors”. Which authors?

We added the reference that was cited above this sentence, to make clear who we were referring to.

Line 366: I would replace “destroyed” with disrupted.

Replaced.

Line 837: GBP5 instead of GBP.  

Fixed this.

Reviewer 2 Report

This review article presents a detailed description on the recent advances in HIV-1 assembly a topic that is very timely and can be useful for a wider scientific community and not only to individuals working on retroviruses.  The review is very well organized and the sections are logically structured. The literature discussed/cited in this article has put together important pieces of information in a very lucid fashion on an intricate process pertaining to the assembly of HIV-1.  The review is straightforward, clear, and overall well-written and is easy to read, despite the relatively descriptive nature of the topic.  Wherever necessary important comparisons of HIV-1 with other retroviruses are also made in an attempt to identify common paradigms.  The summary and outlook for the future is also very well-presented which nicely mentions open questions resulting from the work reviewed in the broader area of HIV-1 assembly.

As mentioned above, this review makes an important contribution to the understanding of HIV-1 assembly and I see no ambiguity in its logical presentation; however, it could be strengthened by including the following suggestions:

In Section 3 (RNA Dimerization and Packaging or in Section 4 on the Central Role of Gag in HIV-1 assembly), it would be important to discuss that specific selection of gRNA over the cellular and spliced RNAs is a multifaceted phenomenon that has been shown to occur in the context of the whole Gag polyprotein, especially in the case of HIV-1 (PMID: 24986025, DOI: 10.1038/ncomms5304; PMID: 27841704, DOI: 10.1080/15476286.2016.1256533; PMID: 29514260, DOI: 10.1093/nar/gky152; PMID: 26237229, DOI: 10.1038/nmeth.3490).  For example, earlier experiments performed using different domains of Gag revealed that the HIV-1 major packaging determinant is primarily confined to stem loop 3 containing a tetra loop (GGAG) which binds to the NC domain of HIV-1 Gag with high affinity (PMID: 9430589, DOI: 10.1126/science.279.5349.384; PMID: 24530126, DOI: 10.1016/j.virol.2014.01.019).  But recent experiments performed using HIV-1 full-length Pr55Gag have demonstrated that a purine-rich internal loop (G//AGG) in SL1 functions as the Pr55Gag binding site and facilitates its gRNA packaging (PMID: 24986025, DOI: 10.1038/ncomms5304).

In Section 11 (Gag Protein Trafficking to the PM or any other place in the article), it would be fitting to discuss the R55W mutant (single amino acid substitution of tryptophan for the arginine residue at position 55) in the matrix of Mason-Pfizer monkey virus (MPMV) described by Rhee and Hunter (PMID: 2170021; DOI: 10.1016/0092-8674(90)90289-q).  This mutant switched the virus particle morphogenesis from intracellular to plasma membrane; therefore, it would be important to discuss whether such a mutant in the matrix domain results in an inherent difference in Gag-RNA binding or is there some other mechanism involved, such as a host factor in this switch?  

Author Response

We thank the reviewer for helpful comments. Responses are indicated below the comments from the reviewer.

In Section 3 (RNA Dimerization and Packaging or in Section 4 on the Central Role of Gag in HIV-1 assembly), it would be important to discuss that specific selection of gRNA over the cellular and spliced RNAs is a multifaceted phenomenon that has been shown to occur in the context of the whole Gag polyprotein, especially in the case of HIV-1 (PMID: 24986025, DOI: 10.1038/ncomms5304; PMID: 27841704, DOI: 10.1080/15476286.2016.1256533; PMID: 29514260, DOI: 10.1093/nar/gky152; PMID: 26237229, DOI: 10.1038/nmeth.3490).  For example, earlier experiments performed using different domains of Gag revealed that the HIV-1 major packaging determinant is primarily confined to stem loop 3 containing a tetra loop (GGAG) which binds to the NC domain of HIV-1 Gag with high affinity (PMID: 9430589, DOI: 10.1126/science.279.5349.384; PMID: 24530126, DOI: 10.1016/j.virol.2014.01.019).  But recent experiments performed using HIV-1 full-length Pr55Gag have demonstrated that a purine-rich internal loop (G//AGG) in SL1 functions as the Pr55Gag binding site and facilitates its gRNA packaging (PMID: 24986025, DOI: 10.1038/ncomms5304).

We have expanded the discussion and citations of this section to include the work outlined by the reviewer. This is now contained in lines 108-117 of the revised version of the manuscript.

In Section 11 (Gag Protein Trafficking to the PM or any other place in the article), it would be fitting to discuss the R55W mutant (single amino acid substitution of tryptophan for the arginine residue at position 55) in the matrix of Mason-Pfizer monkey virus (MPMV) described by Rhee and Hunter (PMID: 2170021; DOI: 10.1016/0092-8674(90)90289-q).  This mutant switched the virus particle morphogenesis from intracellular to plasma membrane; therefore, it would be important to discuss whether such a mutant in the matrix domain results in an inherent difference in Gag-RNA binding or is there some other mechanism involved, such as a host factor in this switch?  

While we did not want to speculate on the mechanisms regulating the R55W switch for M-PMV, we did add this reference to the beginning of the Gag trafficking section to illustrate the importance of MA in determining the site of assembly. This is now on lines 619-623 of the revised manuscript.